# Heightened TLR7/9-Induced IL-10 and CXCL13 Production with Dysregulated NF-ҝB Activation in CD11c^hi^CD11b^+^ Dendritic Cells in NZB/W F1 Mice

**DOI:** 10.3390/ijms20184639

**Published:** 2019-09-19

**Authors:** Lok Yan Yim, Chak Sing Lau, Vera Sau-Fong Chan

**Affiliations:** Division of Rheumatology and Clinical Immunology, Department of Medicine, Li Ka Shing Faculty of Medicine, The University of Hong Kong, Hong Kong 999077, China; adayim@connect.hku.hk

**Keywords:** systemic lupus erythematosus, myeloid dendritic cell, Toll-like receptor 7, Toll-like receptor 9, interleukin 10, C-X-C motif chemokine ligand 13

## Abstract

Systemic lupus erythematosus (SLE) is a chronic, multifactorial autoimmune disease that predominantly affects young females. Dysregulation of different immune cell populations leads to self-tolerance breakdown and subsequent multiple organ damage as the disease develops. Plasmacytoid dendritic cells (pDCs) are potent producers of type I interferon (IFN), while myeloid dendritic cells (mDCs) are more specialized in antigen presentations. We have previously reported that bone-marrow (BM)-derived pDCs from the murine lupus model New Zealand black/white F1 (BWF1) possess abnormalities. Therefore, this study continues to investigate what aberrant properties peripheral pDCs and mDCs possess in BWF1 and how they mediate SLE progression, by comparing their properties in pre-symptomatic and symptomatic mice. Results showed that CD11c^hi^CD11b^+^ myeloid DCs expanded during the disease state with down-regulation of co-stimulatory molecules and major histocompatibility complex class II molecules (MHC II), but their capacity to stimulate T cells was not hampered. During the disease state, this subset of mDCs displayed heightened toll-like receptors 7 and 9 (TLR 7/9) responses with increased interleukin 10 (IL-10) and C-X-C motif chemokine ligand 13 (CXCL13) expressions. Moreover, the expressions of myeloid differentiation primary response 88 (*Myd88*) and nuclear factor kappa B subunit 1 (*Nfkb1*) were higher in CD11c^hi^CD11b^+^ DCs at the disease stage, leading to higher nuclear factor kappa-light-chain-enhancer of activated B cells (NF-κB) p65 phosphorylation activity. In summary, we reported aberrant phenotypic properties with enhanced TLR7/9 responses of CD11c^hi^CD11b^+^ DCs in SLE mediated by aberrant NF-κB signaling pathway. Our findings add additional and novel information to our current understanding of the role of DCs in lupus immunopathogenesis. Lastly, molecular candidates in the NF-κB pathway should be exploited for developing therapeutic targets for SLE.

## 1. Introduction

Systemic lupus erythematosus (SLE) is a chronic, heterogenic, and multifactorial autoimmune disease that causes systemic inflammation and subsequently leads to multiple organ damage. Clinical presentations in SLE patients range from less severe, non-fatal symptoms such as skin lesions, arthritis, and hematological disorders, to life-threatening manifestations, including vital organ failure (e.g., heart, kidney, and lung) [1]. Dysregulated activation of various immune cell populations leading to the breakdown of self-tolerance is one of the major causes of SLE pathogenesis. Dendritic cells (DCs), possessing innate immunity, activate T cells and B cells, which are the major effector cells that cause inflammation and damage in target organs in SLE. There are two major subsets of circulating DCs, namely, plasmacytoid DCs (pDCs) and myeloid DCs (mDCs), each with unique features and functions. Characteristically, pDCs are potent type I interferon (IFN) producers upon viral and bacterial infections [2,3], whereas mDCs are professional antigen-presenting cells with constitutively high expression of major histocompatibility complex class II (MHC II) and costimulatory molecules that allow potent activation of naïve T cells [4]. As such, these two DC subtypes play different roles in SLE pathogenesis [5].

Cellular abnormalities reported in patients have suggested that both pDCs and mDCs might serve as critical mediators in SLE. The up-regulation of IFN-stimulated genes (ISGs) expression in peripheral blood mononuclear cells (PBMCs) and the increase in serum IFNα level in SLE patients were found to be positive correlated with disease activity, implying the functional involvement of pDCs [6,7]. Previously, we and others also reported an abnormal circulating pDC frequency with functional dysregulation, and pDC infiltration was observed in skin lesions and nephritic kidneys of SLE patients [8,9,10,11]. For mDCs, a decreased circulating frequency was reported [11,12] with prominent infiltration in nephritic kidneys [9]. Functionally, monocyte-derived mDCs (mo-mDCs) from SLE patients also displayed increased CD80, CD86, and MHC II expressions with enhanced T cell stimulating ability and interleukin (IL)-8 production [13]. These findings suggest that both pDCs and mDCs may mediate disease pathogenesis via aberrant functional properties.

DCs express various toll-like receptors (TLRs) that sense endogenous danger signals via binding to damage-associated molecular patterns (DAMPs) in the context of autoimmunity. TLR7 and TLR9 are of particular importance in SLE, as they, respectively, detect RNA- and DNA-containing autoantigens [14,15]. In humans, pDCs highly express TLR7 and TLR9 but expressions of these receptors are not detected in mDCs. In contrast, both pDCs and mDCs in mice express TLR7 and TLR9 [16]. In SLE pathogenesis, the increase in apoptosis and the insufficient clearance of apoptotic materials cause the accumulation of autoantigens including nucleic acids [17,18,19], which in turn may contribute to the breakdown of self-tolerance via TLRs stimulation. Chromatin-containing microparticles derived from sera of SLE patients have been shown to up-regulate co-stimulatory molecules expressions with increased production of IL-6 and tumor-necrosis factor alpha (TNFα) in mDCs, and enhance IFNα production in pDCs in vitro [20]. Moreover, RNA-containing immune complexes derived from patients have also been shown to activate bone marrow (BM)-derived mDCs to produce IFNα and IL-6 in C57BL/6 mice [15]. Upon TLR7/9 stimulation, the adaptor molecule myeloid differentiation primary response 88 (MyD88) is recruited and complexed with other signaling molecules such as the interferon regulatory factor (IRF) 5, tumor-necrosis factor (TNF)-receptor-associated factor (TRAF) 3, TRAF6, and interleukin-1-receptor-associated kinase 4 (IRAK4). The subsequent downstream activation of IRF7 and nuclear factor kappa–light–chain-enhancer of activated B cells (NF-κB) initiate type I IFN and proinflammatory cytokines transcription respectively [2,21].

Although studies using SLE patient samples have revealed aberrations in frequency, phenotypes, and functions in both DC subtypes, some of the reported findings were not consistent [9,11]. Likely, confounding variables such as medications, lifestyle variations, and environmental factors may have contributed to the discrepancy. Moreover, patients only seek medical treatment after symptoms develop; thus, it is not possible to follow the changes in DC properties before disease onset in order to fully evaluate how different DC phenotypes contribute to disease progression. Therefore, this study aims to identify abnormalities that pDCs and mDCs develop in disease onset and to study how these abnormalities meditate SLE development using the murine NZ black/white F1 (BWF1) lupus model. This model is chosen because of its high resemblance to human disease in terms of female predisposition, disease heterogeneity, and type I IFN-dependence [22,23]. By systematically evaluating the abundance and phenotypic and functional properties of DCs before and after disease onsets in BWF1, we show that CD11c^hi^CD11b^+^ DCs exhibit enhanced TLR7- and TLR9-mediated production of IL-10 and C-X-C motif chemokine ligand 13 (CXCL13) as the disease progresses, and mechanistically, this is associated with dysregulation in the NF-κB signaling pathway.

## 2. Results

### 2.1. Expanded CD11c^hi^CD11b^+^ Dendritic Cells Display Aberrant Phenotypes during SLE Development

To evaluate the properties of splenic DCs in BWF1 mice, we first used CD11c, B220, CD317, and Siglec-H to distinguish pDCs [24,25]. After comparing different gating strategies, we found that pDCs were best identified as CD11c^dim^CD317^+^ cells, which constituted a distinct population of B220^+^Siglec-H^+^ cells (Figure 1A). Similarly, using the common mDCs markers CD11c and CD11b [26], a distinct population of CD11c^hi^CD11b^+^ cells constitutively expressing high levels of MHC class II and the costimulatory molecule CD80 could be identified (Figure 1B). Using these markers, the frequency and total number of pDC and CD11c^hi^CD11b^+^ mDC were compared in BWF1 mice before and after symptomatic SLE onset, which was marked by an increase in anti-dsDNA antibody level and the manifestation of proteinuria [27]. Results showed that there was a lower splenic pDCs frequency in symptomatic (Sym) mice when compared with pre-symptomatic (Pre-Sym) mice (0.48 ± 0.04% VS. 1.15 ± 0.06%, *p* ≤ 0.0001) (Figure 1C), while this difference was not observed in age- and sex-matched non-lupic parental NZW strain (Appendix A). However, there was no change in total number of pDCs (Figure 1C), and the reduced frequency was likely due to the increase in splenic cellularity during disease progression in F1 mice (Appendix A). In contrast, there was an increase in both the frequency (Sym VS. Pre-sym, 1.37 ± 0.21% VS. 0.81 ± 0.07%, *p* ≤ 0.05) and total number (2.20 ± 0.49 × 10^6^ VS. 0.60 ± 0.05 × 10^6^, *p* ≤ 0.01) of CD11c^hi^CD11b^+^ DCs in symptomatic mice (Figure 1D) but not in NZW controls (Appendix A), suggesting that the increase in CD11c^hi^CD11b^+^ DC abundance in symptomatic F1 mice may not be due to age difference but likely related to the development of SLE.

Since splenic CD11c^hi^CD11b^+^ DCs in symptomatic BWF1 mice expended, we hypothesized that this population may acquire aberrant phenotypic and functional properties to facilitate disease development. In the context of professional antigen-presenting cells for T cell activation, the expressions of different co-stimulatory molecules and MHC II in CD11c^hi^CD11b^+^ DCs were evaluated. Surprisingly, as F1 mice progressed from pre-symptomatic to the symptomatic stage, CD11c^hi^CD11b^+^ DCs expressed lower levels of CD40 (MFI: 515.3 ± 24.12 VS. 409.0 ± 19.22, *p* ≤ 0.01), CD80 (1530.0 ± 156.5 VS. 677.6 ± 62.24, *p* ≤ 0.001) and MHC II (13,317 ± 3733 VS. 4682 ± 1671, *p* ≤ 0.05) (Figure 2A,B). The frequencies of MHC II (95.70 ± 0.48% VS. 60.99 ± 6.18%, *p* ≤ 0.001) and CD80 (87.40 ± 2.41% VS. 39.47 ± 6.15%, *p* ≤ 0.0001) expressing CD11c^hi^CD11b^+^ DCs in symptomatic mice were also dramatically reduced. In the non-lupic NZW controls, no marked difference in these parameters was observed as the mice aged (Appendix A). The reduced MHC II and costimulatory molecules expressions in F1 mDCs from symptomatic mice, however, did not appear to have significant impact on their capacity to stimulate allogeneic T cell proliferation, at least in in vitro with a DC/T ratio of 1:10 (Figure 2C). Whether a functional difference can be observed at a lower DC/T ratio is yet to be determined.

### 2.2. CD11c^hi^CD11b^+^ Dendritic Cells Display Heightened TLR7/9 Response during Disease State

As murine mDCs also sense nucleic acids-containing autoantigens via TLR7 and TLR9, we next measured the expressions of various inflammatory and lymphocyte-stimulating cytokines upon R837 (TLR7 ligand) and CpG (TLR9 ligand) stimulation in vitro. Recently, the role of follicular helper T cell (Tfh) in mediating SLE pathogenesis has drawn great attention because Tfh are crucial in facilitating B cell activation and germinal center (GC) development [28]. The ability of CD11c^hi^CD11b^+^ DCs to produce the Tfh attracting chemokine CXCL13 was also evaluated. Results showed that apart from *Il-6*, unstimulated CD11c^hi^CD11b^+^ DCs from symptomatic mice generally had higher basal mRNA expressions of *Tnfa*, B-cell activating factor (*Baff*), *Il-10,* and *Cxcl13* than pre-symptomatic mice (Figure 3A). Again, such difference was not observed in the NZW controls (Appendix A). This suggests that CD11c^hi^CD11b^+^ DCs may be more prone to produce these cytokines and chemokines upon stimulation in the disease state. Indeed, TLR7-induced *Cxcl13* (6.49 ± 1.84 VS. 0.74 ± 0.12, *p* ≤ 0.05), and TLR9-induced *Cxcl13* (12.6 ± 0.28 VS. 0.71 ± 0.28, *p* ≤ 0.05) and *Il-10* (43.8 ± 10.7 VS. 5.89 ± 1.61, *p* ≤ 0.01) mRNA expressions in CD11c^hi^CD11b^+^ DCs from symptomatic mice, were significantly higher than pre-symptomatic mice (Figure 3B). In contrast, TLR4 stimulation of CD11c^hi^CD11b^+^ DCs by LPS did not show any differences in *Il10* and *Cxcl13* induction (Appendix A), indicating that heightened TLR responses were specific to TLR7 and TLR9 stimulation. Consistently, these DCs from symptomatic mice produced more IL-10 and CXCL13 protein upon TLR7 or TLR9 stimulation, whereas these factors were barely detected in its pre-symptomatic counterparts (Figure 4).

### 2.3. Enhanced TLR7/9 Signaling in CD11c^hi^CD11b^+^ in F1 Mice at Disease State

To dissect the underlying mechanism of the heightened TLR7 and TLR9 responses in symptomatic BWF1 CD11c^hi^CD11b^+^ DCs to produce IL-10 and CXCL13, we hypothesized that the aberrant expressions of these receptors might be involved. However, results showed that both TLR7 and TLR9 expression in pre-symptomatic and symptomatic mice revealed no significant difference, or even a mild decrease in frequency of TLR9-expressing CD11c^hi^CD11b^+^ DCs in symptomatic mice (Figure 5). We next suspected that there might be abnormalities in signaling transduction downstream of TLR7 and TLR9. We evaluated the expressions of MyD88, the key adaptor molecule recruited to TLRs for transducing intracellular signals; TRAF3, which is essential for induction of IL-10 expression [29]; IRF5, a novel regulator of CXCL13 expression [30]; and nuclear factor kappa B subunit 1 (*Nfkb1*), a key component of the canonical NF-κB transcription factor for inflammatory responses [31]. Results showed that the expressions of both *Myd88* and *Nfkb1* in CD11c^hi^CD11b^+^ DCs were significantly higher in symptomatic than pre-symptomatic mice, while no difference was noted in *Traf3* and *Irf5* expressions (Figure 6A–D). Nevertheless, TLR-mediated NF-κB activity could be dysregulated. Indeed, CD11c^hi^CD11b^+^ DCs from symptomatic mice exhibited higher CpG-induced phosphorylation of the NF-κB p65, while minimal phosphorylation was observed when compared with pre-symptomatic mice, as evaluated by Western blot (Figure 6E) and flow cytometry analysis (Appendix A).

## 3. Discussion

Collectively, we demonstrated that splenic CD11c^hi^CD11b^+^ DCs in BWF1 mice expanded with aberrant phenotypic and functional properties during SLE development. As the disease progressed, CD11c^hi^CD11b^+^ DCs showed reduced co-stimulatory molecules and MHC II expressions without reducing their T cell stimulation ability in vitro. Moreover, these splenic DCs in disease mice displayed heightened TLR7- and TLR9-mediated responses, as indicated by the elevated expressions of the B cell-stimulating cytokine IL-10 and chemoattractant CXCL13, with the latter also being critical for Tfh migration into B cell follicles for GC formation during lupus pathogenesis. In the last part of our study, we tried to dissect the mechanism of elevated TLR7 and TLR9 responses in CD11c^hi^CD11b^+^ DCs and found that this might possibly be due to the increased expression of the adaptor protein MyD88 and activation of the transcription factor NF-κB. Our findings have suggested mechanism that may help to unwind how DCs facilitate SLE pathogenesis via aberrant TLR7 and TLR9 signaling upon stimulation by accumulated nucleic acids.

Our observation on the expansion of splenic CD11c^hi^CD11b^+^ DCs in BWF1 mice during disease state is consistent with a previous study by Gleisner et al. [24], in which they defined cDC as CD11c^hi^B220^-^ cells. These cDCs expanded in peripheral blood, bone marrow, and lymph nodes of BWF1 lupus mice, and similar observation was reported in CD11c^+^CD11b^+^ DCs in B6.*Sle1.Sle2.Sle3* [24,32]. Apparently, the expansion of DC facilitates autoimmune responses via priming of proinflammatory T cell subsets [33], inducing memory B cell differentiation and potentiating autoantibody production as reported in various lupus mouse models, including B6.NZBc1 and MRL. *Fas^lpr^* [34,35]. In luic BWF1 mice, this enhancement was suggested to be linked to the over-expression of costimulatory molecules like CD80 and CD86 in B220^-^CD11c^hi^ cDCs [24]. In contrast, we observed a dramatic reduction of CD80 and MHC II expression on CD11c^hi^CD11b^+^ DCs from symptomatic BWF1. Study by Wan et al. also reported that splenic and lymph node CD11c^hi^CD11b^+^ DCs from B6.*Sle1.Sle2.Sle3* congenic mice expressed a lower level of CD80 when compared with wild-type C57/BL6 mice [36]. Interestingly, CD11c^hi^ DCs with low MHC II expression were found in BWF1 kidneys only after proteinuria onset [37]. The use of different mDC markers in different lupus models may partially account for the discrepancy in co-stimulatory molecules expression between these studies. In addition, the physiological state at which the mDC phenotypes were evaluated in different studies might also contribute to the difference. As a mechanism to control immune responses after activation, antigen-loaded mDCs would have surface MHC II molecules degraded upon encountering antigen-specific T cells [38]. Since we examined symptomatic BWF1 mice after proteinuria onset, i.e., mid-late lupus disease state, likely the lower expressions of co-stimulatory molecules and MHC II on mDCs might be a consequence of mDC activation after T cell priming during SLE development, and the early but transient upregulation of expressions of these molecules cannot be excluded. Despite the reduced expression of costimulatory and MHC II molecules, CD11c^hi^CD11b^+^ DCs from the symptomatic mice still retained comparable allogeneic T cell stimulation capacity in vitro.

Given that TLR7 and TLR9 are the key receptors for sensing RNA or DNA-containing immune complexes in SLE, delineating the response of mDCs upon stimulation of these receptors may reveal how mDCs mediate SLE pathogenesis. We showed that CD11c^hi^CD11b^+^ DCs displayed heightened TLR7 and TLR9 responses with increased IL-10 and CXCL13 production in symptomatic BWF1 mice. IL-10 is a pleotropic cytokine that stimulates B cells to produce antibodies, but at the same time also functions as a regulatory cytokine that inhibits T cell responses. Because of the dual function of IL-10, its role in SLE is rather complex, and discrepant findings have been reported. In an early study, the continuous administration of anti-IL-10 antibody may have delayed SLE onset in BWF1 mice [39]. However, IL-10 deficiency in MRL.*Fas^lpr^* mice led to disease exacerbation with more severe glomerulonephritis and higher mortality [40]. In B6.*Sle1.Sel2.Sl23* congenic mice, the continuous overexpression of low levels of IL-10 also delayed autoantibody production and clinical nephritis [41]. It is possible that different mouse strains could have different thresholds for IL-10 level, and thus would have different outcomes in modulating IL-10 level. In MRL.*Fas^lpr^* mice, a complete deficiency in IL-10 was accompanied by an enhanced IFNγ production in both CD4^+^ and CD8^+^ T cells, and an elevated anti-dsDNA antibody production with more severe clinical diseases, suggesting a protective role of IL-10 in lupus development [40]. However, in SLE patients, a higher serum level of IL-10 was found to be predictive of higher disease activity in subsequent clinic visits [42], suggesting a pathogenic role for IL-10. Therefore, the increased IL-10 production by mDCs via enhanced TLR7 and TLR9 responses could play different roles during the course of SLE pathogenesis.

One of the key novel findings of this study is the increased production of CXCL13 via heightened TLR7 and TLR9 responses (Figure 4). CXCL13 is a chemoattractant that promotes migration of CXCR5 expressing cells. DC-derived CXCL13 is critical for splenic B cell migration, as the ablation of follicular DCs has led to the disruption of germinal center [43]. As such, DC-derived CXCL13 also attracts CXCR5^+^ Tfh cells that play an important role in facilitating B cell activation and germinal center development [28]. Clinically, serum level of CXCL13 was found higher in SLE patients and in positive correlation with disease activity [44,45], and Tfh was shown to facilitate GC reactions, thereby contributing to lupus pathogenesis in different murine models [46,47]. It is possible that mDCs increase CXCL13 production via heightened TLR 7 and 9 responses to facilitate Tfh migration and GC reaction during disease development. Although previous study reported that splenic and thymic CD11c^+^CD11b^+^ DCs from symptomatic BWF1 express CXCL13 mRNA [48], and blood and bone marrow (BM)-derived CD11c^+^ DCs also increase CXCL13 mRNA expression when stimulated by the proinflammatory cytokine TNFα [32], how DCs response to TLR stimulation in terms of CXCL13 production has not been evaluated in SLE. Here, we demonstrated that increased production of CXCL13 by CD11c^hi^CD11b^+^ DCs in the disease state was due to increased TLR7 and TLR9 responses, and that the enhanced NF-κB activation might possibly serve as one of the underlying pathological mechanisms. NF-κB pathway is involved in the transcription of various proinflammatory cytokines in SLE [49], and MyD88 and NF-κB are the adaptor molecule and transcriptional factor respectively within the TLR7 and TLR9 signaling pathway. Therefore, increased expression of MyD88 in CD11c^hi^CD11b^+^ DCs observed in this study may possibly increase the transcription of proinflammatory cytokines via TLR7 and TLR9 stimulation by RNA or DNA-containing autoantigens. Additionally, specific deletion of MyD88 in DCs prevented the development of dermatitis with decrease expression of the proinflammatory cytokines IL-6 and IL-12, and different B cell-stimulating cytokines, including IL-10 with hampered B cell proliferation in MRL.*Fas^lpr^* mice [50].

In human and murine lupus nephritic tissues, enhanced NF-κB activation was also detected, and this could drive the proliferation of mesangial cells (a primary lesion characteristics of lupus nephritis) and expression of a range of chemotactic factors, including CCL2, CCL5, MCP-1, and IL-8, in vitro [51,52]. Moreover, systemic inhibition of NF-κB signaling by specific chemical inhibitor resulted in amelioration of nephritis in MRL.*Fas^lpr^* with decreased cytokine production and improved survival [52]. Recently, it has been reported that inhibition of the NF-κB-inducing kinase (NIK) of the non-canonical NF-κB pathway in murine and human monocyte-derived DCs can decrease IL-12p40 production in vitro, and global inhibition of NIK in BWF1 can ameliorate SLE manifestations [53]. These studies highlight the role of NF-κB signaling pathway in mediating disease development; and our findings further suggest that the increased activation of NF-κB is, at least in part, responsible for the dysregulated TLR7 and TLR9 responses, thereby leading to the hyper-production of IL-10 and CXCL13 in CD11c^hi^CD11b^+^ DCs at the disease stage.

Murine CD11c^hi^CD11b^+^ or CD11c^+^CD11b^+^ DCs were regarded as mDCs in autoimmune research [36,54]. Myeloid DCs can be sub-divided into monocyte-derived DCs (MoDCs) and classical/conventional DCs (cDCs), the latter of which most efficiently primes T cell activation. Further studies have also revealed that cDCs can be classified into two major subsets of cDC1 and cDC2, with distinct phenotypes and functions. cDC1 is more efficient in cross-presentation of antigen to prime CD8 T cell response as well as Th1 response, whereas cDC2 contributes to regulation of Th2 and Th17 responses. Recent advancements in high-dimensional immunophenotyping using mass-cytometry have greatly increased the number of detection parameters, thus allowing a more detailed immune cells characterization and classification via the inclusion of additional markers. An elegant study by Guilliams et al. using unsupervised clustering analysis of mass cytometry data demonstrated that cDC1 and cDC2 could be distinguished based on differential expression of XCR1, CD24, CD172a, and CD11b, in addition to high CD11c and CD26 expressions [55]. Interestingly, kinetic changes in the abundance of these cDC subsets were also reported, with decreased abundance of cDC1 and increased for 2 cDC2 subtypes during the course of inflammation [55]. In this respect, our pilot data showed that the CD11c^hi^CD11b^+^ DCs in spleen of BWF1 mice were mainly cDC2 that expressed CD172a but not the cDC1 marker Clec9a, and their frequency did not show significant fluctuation as the mice developed lupus (Appendix A). The reduced MHC II and CD80 expression were thus associated with the cDC2 cells (Appendix A). To date, there is no study delineating the role of cDC1 and cDC2 in lupus. Our current study can be further elaborated to dissect how different cDC populations are dysregulated during SLE pathogenesis by inclusion of other DC markers described above.

## 4. Materials and Methods

### 4.1. Mouse Colonies and Disease Monitoring

NZB/BINJ and NZW/LacJ were purchased from The Jackson Laboratory (Bar Harbor, ME, USA). These lines were maintained and interbred to generate F1 generation (BWF1) in the Minimal Disease Area of the Laboratory Animal Unit, The University of Hong Kong. Disease development in BWF1 was assessed by monitoring serum level of anti-dsDNA IgG and development of proteinuria, as previously described [27]. Mice between 8 and15 weeks old without an increase in anti-dsDNA IgG (level comparable to the non-lupus parental strain NZW) and negative for proteinuria were defined as pre-symptomatic mice, while mice above 20 weeks old with elevation of anti-dsDNA IgG (level higher than two standard deviations of mean serum level in NZW) and proteinuria development (3mg/mL or above) for at least two consecutive weeks were defined as symptomatic mice. Age- and sex-matched non-lupus parental strain NZW was used as controls in specified experiments (young NZW between 11 and 15 weeks old, old NZW between 30 and 40 weeks old). All animal experiments were approved by the Committee on the Use of Live Animal in Teaching and Research (CULATR, approval number 3412-14 and 3647-15), the University of Hong Kong.

### 4.2. Antibodies for Immunophenotyping

Antibodies used for flow cytometry analysis or FACS purification were all monoclonal antibodies purchased from BD Biosciences (San Jose, CA, USA), eBioscience (San Diego, CA, USA), or BioLegend (San Diego, CA, USA). The following antibodies were used: anti-CD11c-FITC (clone HL3), anti-TLR7-PE (clone A94B10), anti-TLR9-PE (clone J15A7), anti-CD40-PE (clone 3/23), anti-CD80-PE (clone 1610A1), anti-Siglec-H-PE (clone 551) anti-CD317-APC (clone eBio927), anti-B220-PE-Cy7 (clone RA3-6B2), anti-MHC II-PE-Cy7 (clone M5/114.152), and anti-CD11b-biotin (clone M1/70). Streptavidin-PE-Cy7 was used as secondary detection agent where appropriate. Surface staining was performed in PBS containing 2% heat-inactivated fetal bovine serum (FBS), 1mM EDTA, and 0.05% sodium azide, at 4 °C in dark. For intracellular staining of TLR7 and TLR9, cells were first surface stained and fixed in 4% paraformaldehyde, followed by intracellular staining in 0.2% saponin. Flow cytometry was done using FC500 (Beckman Coulter, Brea, CA, USA), and data were analyzed by FlowJo software (Treestar Inc., Ashland, Ore, USA)

### 4.3. Purification of and Stimulation of CD11c^hi^CD11b^+^ DCs

Spleens were meshed mechanistically to generate single cell suspensions, and red blood cells were lysed using ACK buffer. T and B cells were first depleted by magnetic-activated cell sorting (MACS) (Miltenyi, Bergisch Gladbach, Germany) using anti-CD3 (clone 145-2C11) and anti-CD19 (clone 1D3) antibodies to enrich DCs. The DC-enriched fraction was subsequently stained with anti-CD11c and anti-CD11b and purified by fluorescence-activated cell sorting (FACS). Class C CpG (ODN 2395, 1 μM,) and R837 (Imiquimod, 2.5 ug/mL) (both from InvivoGen, San Diego, CA, USA) were used for TLR9 and TLR7 stimulation, respectively, when evaluating inductions of cytokines and chemokines expressions in mDCs in vitro. For evaluating NF-ҝB activation in CD11c^hi^CD11b^+^ DCs, 5 μM class C CpG was used. Purified CD11c^hi^CD11b^+^ DCs were stimulated in complete IMDM medium with supplements of 20 mM GlutaMAX, 1 mM sodium pyruvate, 50 μM beta-mercaptoethanol, non-essential amino acids, 100 U/mL penicillin, 100 μg/mL streptomycin, and 10% FBS (all from Gibco, Carlsbad, CA, USA), at a density of 1 × 10 ^5^ cells/well in 200 μL for 5 h, to evaluate mRNA induction. For detection of cytokine and chemokine production, mDCs were stimulated for 24 h.

### 4.4. Allogeneic T Cell Proliferation Assay

Purified BWF1 CD11c^hi^CD11b^+^ DCs were irradiated by 3000 cGy of gamma irradiation to prevent cell proliferation. T cells were prepared from C57BL/6 splenocytes by negative depletion of CD19^+^ B cells (CD3^+^ T cell purity ≥ 95%). A total of 1 × 10^5^ C57BL/6 T cells were co-cultured with CD11c^hi^CD11b^+^ DCs in 10:1 ratio in 200 µL complete IMDM for 2 days, and 0.5 μCi ^3^H-thymidine was added to each well and further cultured for 24 h. After incubation, cells were harvested to the UniFilter (Perkin Elmer, Waltham, Mass, USA), washed with deionized water, and dried overnight at 50 °C. Scintillant was then added and the UniFilter was read by the TopCount NXT Microplate Scintillation and Luminescence Counter (Perkin Elmer, Waltham, Mass, USA).

### 4.5. Quantitative RT-PCR

Cells were harvested in Tri Reagent^®^ solution (Invitrogen, Thermo Fisher Scientific, Waltham, MA, USA), and RNA was extracted according to manufacturer’s instructions. Reverse transcription was performed using ThermoScript™ RT-PCR Systems (Life Technology, Thermo Fisher Scientific, Waltham, MA, USA) followed by qPCR using the KAPA SYBR^®^ FAST qPCR Kit (Sigma-Aldrich, St. Louis, Mo, USA). The qPCR was performed using StepOnePlus Real-Time PCR system (Invitrogen, Thermo Fisher Scientific, Waltham, MA, USA) with the following thermal condition: 95 °C for 4 min, 40 cycles of 95 °C, 62 °C, and 72 °C for 20 s each. The mRNA expression was normalized to the house-keeping gene *β-actin* and expressed as relative quantity (RQ) calculated by the formula 2^−^[ΔCq (test sample) − ΔCq (reference sample)]. The primer sequences are listed in Table 1.

### 4.6. ELISA

ELISA detection for IL-10 and CXCL13 (BioLegend, San Diego, CA, USA) was performed according to the procedures recommended by the manufacturer. Briefly, capture antibody pre-coated 96-well-plate was blocked with 1% BSA in PBS before addition of 50 μL of samples or standard. The plate was then incubated for 2 h at room temperature, followed by 1 h incubation with biotinylated detection antibody, and 30 min incubation with streptavidin-conjugated horseradish peroxidase (HRP). Afterwards, 3,3′5,5′tetramethylbenzidine (TMB) (BD, San Jose, CA, USA) substrate was added and the reaction was stopped by adding equal volume of 2N sulfuric acid. The plate was then read by plate-reader (BMG Lab Tech, Offenburg, Germany) at 450 nm with 630 nm as the correction wavelength.

### 4.7. Immunoblotting

Cells were lysed in fresh cell lysis buffer (Cell Signaling Technology, Boston, MA, USA) containing 1 μM PMSF, and HALT protease and phosphatase inhibitors (Thermo Scientific, Carlsbad, CA, USA). Twenty micrograms of protein from each sample was resolved on a 12.5% SDS-PAGE and transferred to a polyvinylidene difluoride membrane. After blocking with 5% blocking reagent (BioRad, Hercules, CA, USA) in Tris-buffered saline with 0.5% Tween-20 (TBS-T), the membrane was incubated overnight at 4 °C with either anti-NF-κB p65 (C22B4), anti-phospho-NF-κB p65 (Ser536), or β-actin (8H10D10) primary antibody (Cell Signaling Technology, Boston, MA, USA), followed by incubation with HRP-conjugated anti-rabbit secondary antibody for 2 h at room temperature. Signals were then detected by exposure to X-ray films after the addition of ECL substrate. Densitometry analysis was done using the ImageJ v1.47 (National Institutes of Health, Bethesda, Md, USA) software.

### 4.8. Statistical Analysis

Unpaired two-tail Student’s t-test was performed using GraphPad Prism (GraphPad, Inc., San Diego, CA, USA). Data presented were from at least three independent experiments, unless specified.

## 5. Conclusions

In summary, the current study reported an expansion of CD11c^hi^CD11b^+^ DCs in nephritic stage of lupus in BWF1 mice. These DCs displayed a heightened TLR7 and TLR9 response, with increased expression of IL-10 and CXCL13. Moreover, the increase in TLR7 and TLR9 response in CD11c^hi^CD11b^+^ DCs was associated with elevated expressions of *Myd88* and *Nfkb1* and enhanced canonical NF-κB phosphorylation. Our study highlighted that CD11c^hi^CD11b^+^ DCs possess functional abnormalities during SLE development and provided new insights into the possible role of mDCs in facilitating Tfh migration into germinal center to potentiate B cell responses via heightened TLR7 and TLR9 response upon stimulation of accumulated nucleic acids-containing immune complex during SLE development.

## Figures and Tables

**Figure 1 ijms-20-04639-f001:**
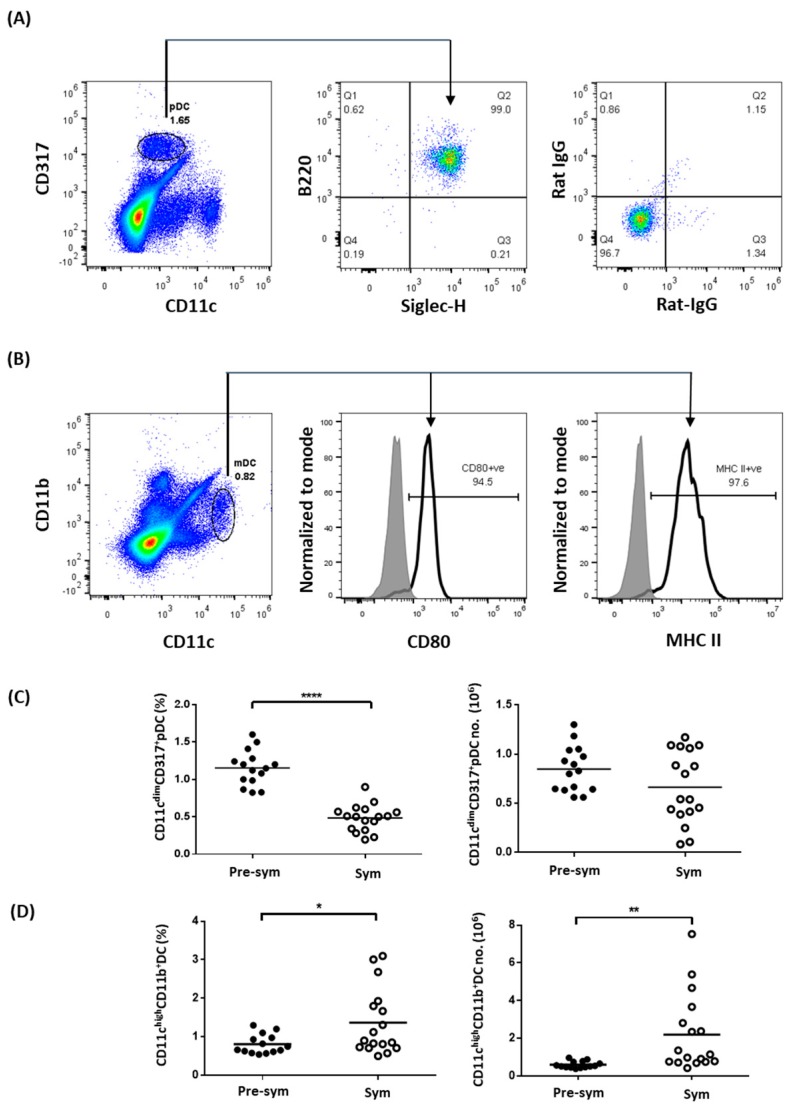
The abundance of myeloid dendritic cells (mDCs) but not plasmacytoid dendritic cells (pDCs) increases in symptomatic black/white F1 (BWF1). Total splenocytes were stained with different pDC and mDC markers to distinguish the two dendritic cell (DC) subtypes in the spleen of BWF1 using flow cytometry. (**A**) Splenocytes were stained with the pDC markers CD11c, CD317, B220, and Siglec-H. Expression of B220 and Siglec-H were detected on CD11c^dim^CD317^+^ gated cells (indicated by arrow). The right panel represents isotype control antibodies staining. (**B**) Splenocytes were stained for mDC markers CD11c and expressions of CD80 and MHC II were evaluated within the CD11c^hi^CD11b^+^ gated population (indicated by arrows). (**C**) Summary plots comparing the frequency and total number of CD11c^dim^CD317^+^pDCs and (**D**) CD11c^hi^CD11b^+^ DCs from pre-symptomatic (pre-sym) and symptomatic (sym) mice. Each symbol represents an individual mouse, and student’s *t*-test was used for statistical analysis (* *p* ≤ 0.05, ** *p* ≤ 0.01, **** *p* ≤ 0.0001).

**Figure 2 ijms-20-04639-f002:**
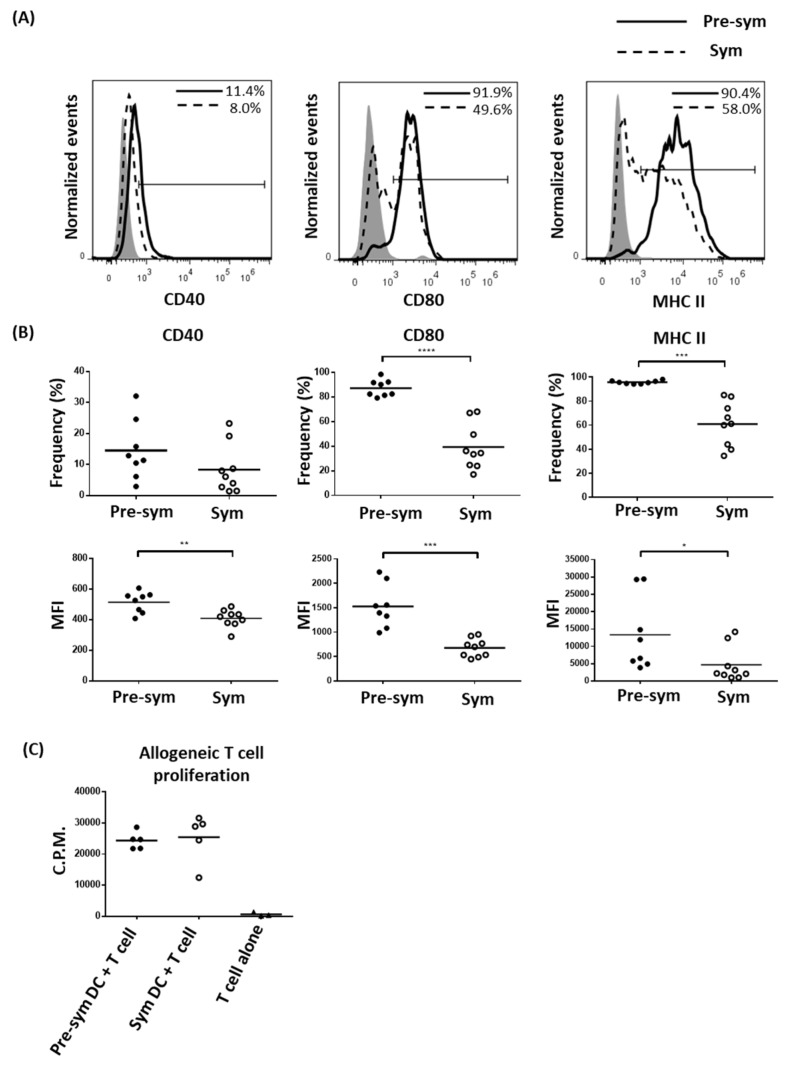
Dampen expressions of co-stimulatory molecules and MHC II on symptomatic BWF1 CD11c^hi^CD11b^+^ DCs does not hamper the ability of mDCs to induce T cell proliferation. Splenocytes from pre-symptomatic and symptomatic black/white F1 (BWF1) were isolated and stained for CD11c and CD11b that are also mDC markers together with the indicated activation marker. (**A**) Representative histograms showing the expression of different activation markers on CD11c^hi^CD11b^+^ DCs from pre-symptomatic (solid line) and symptomatic BWF1 (dotted line), respectively. Shaded histogram represents the isotype control. (**B**) Summary plots comparing the expression of CD40, CD80, and MHC II on CD11c^hi^CD11b^+^ DCs from pre-sym and sym BWF1 in terms of frequency (%) and mean fluorescence intensity (MFI). Each symbol represents an individual mouse. (**C**) Fluorescence-activated cell sorting (FACS)-purified splenic CD11c^hi^CD11b^+^mDCs were co-cultured with C57BL/6 T cells in 1:10 ratio for two days. 3H-thymidine was then added and cultured for another 24 h to evaluate T cell proliferation using thymidine incorporation assay. A summary plot comparing the ability of mDCs from pre-symptomatic (filled circle) and symptomatic mice (open circle) to induce allogeneic T cell proliferation is shown, and each symbol represents the mean counts per minute (C.P.M.) of triplicate wells of an individual mouse. For (**A**) to (**C**), student’s t-test was used for statistical analysis (* *p* ≤ 0.05, ** *p* ≤ 0.01, *** *p* ≤ 0.001, and **** *p* ≤ 0.0001).

**Figure 3 ijms-20-04639-f003:**
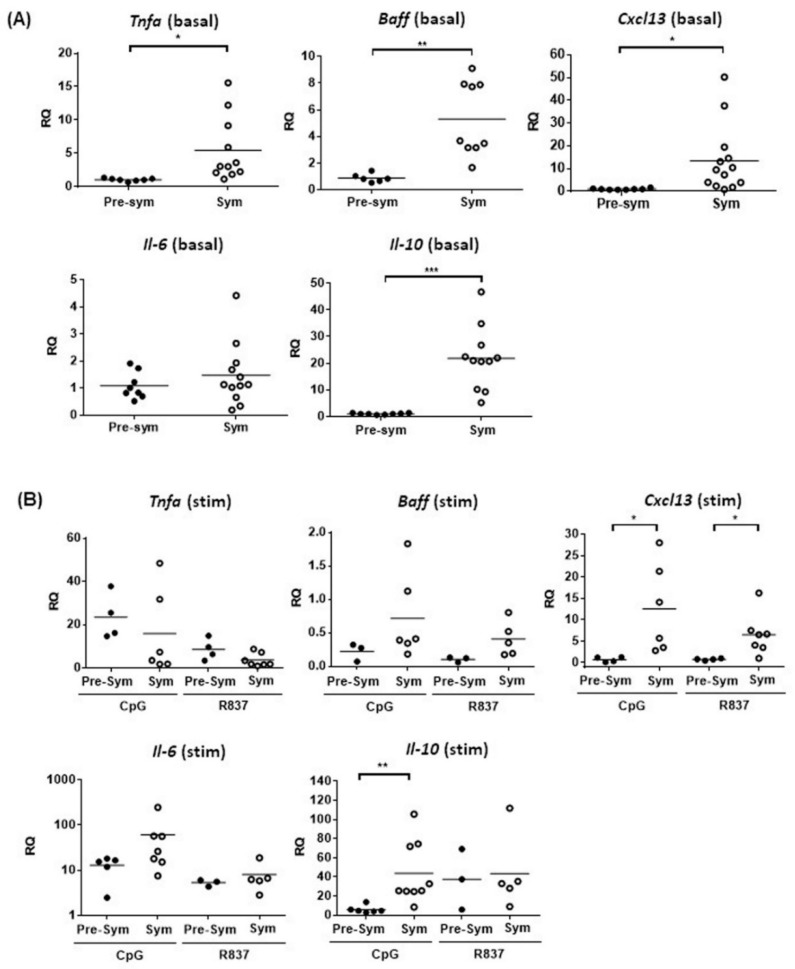
The basal expressions of different cytokines and chemokines in BWF1 mDCs are higher in symptomatic mice with CD11c^hi^CD11b^+^ DCs displaying a heightened TLR7/9 response. Splenic CD11c^hi^CD11b^+^ DCs were FACS-purified, and mRNA expressions of *Tnfa*, *Baff*, *Il-6*, *Il-10,* and *Cxcl13* was determined by quantitative polymerase chain reaction (qPCR). The mRNA expression of each target was normalized with the house-keeping gene *β-actin*. (**A**) Summary plot showing the basal mRNA expression of the indicated target in CD11c^hi^CD11b^+^ DCs from pre-sym and sym BWF1. The relative quantity (RQ) was the level of mRNA relative to unstimulated CD11c^hi^CD11b^+^ DCs from pre-symptomatic mice. (**B**) The mRNA induction of the indicated target in CD11c^hi^CD11b^+^ DCs from pre-symptomatic and symptomatic BWF1 upon toll-like receptor (TLR) stimulation. Cells were stimulated for 5 h in the presence or absence of the TLR9 ligand CpG (1 µM) or TLR7 ligand R837 (2.5 μg/mL). The induction of mRNA is expressed as relative quantity (RQ) relative to un-stimulated CD11c^hi^CD11b^+^ DCs of the respective group. For (A) and (B), each symbol represents an individual mouse, and student’s *t*-test was used for statistical analysis (* *p* ≤ 0.05, ** *p* ≤ 0.01, and *** *p* ≤ 0.001).

**Figure 4 ijms-20-04639-f004:**
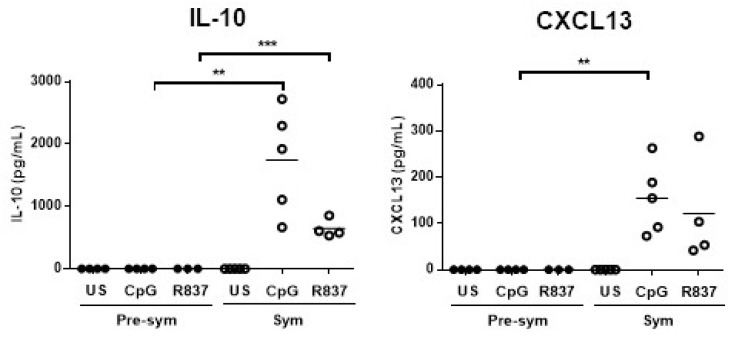
Symptomatic BWF1 CD11c^hi^CD11b^+^ DCs are potent in producing IL-10 and CXCL13, while their pre-symptomatic counterpart does not produce these cytokines upon TLR7/9 stimulation in vitro. Summary plots comparing IL-10 and CXCL13 protein production by TLR7- or TLR9-stimulated CD11c^hi^CD11b^+^ DCs from pre-sym and sym BWF1. FACS-purified CD11c^hi^CD11b^+^ DCs were stimulated with the TLR9 ligand CpG (1μM) or TLR7 ligand R837 (2.5 μg/mL) for 24 h. ELISA was used to detect the indicated cytokine in supernatant and each symbol represents an individual mouse. Student’s t-test was used for statistical analysis (** *p* ≤ 0.01, *** *p* ≤ 0.001).

**Figure 5 ijms-20-04639-f005:**
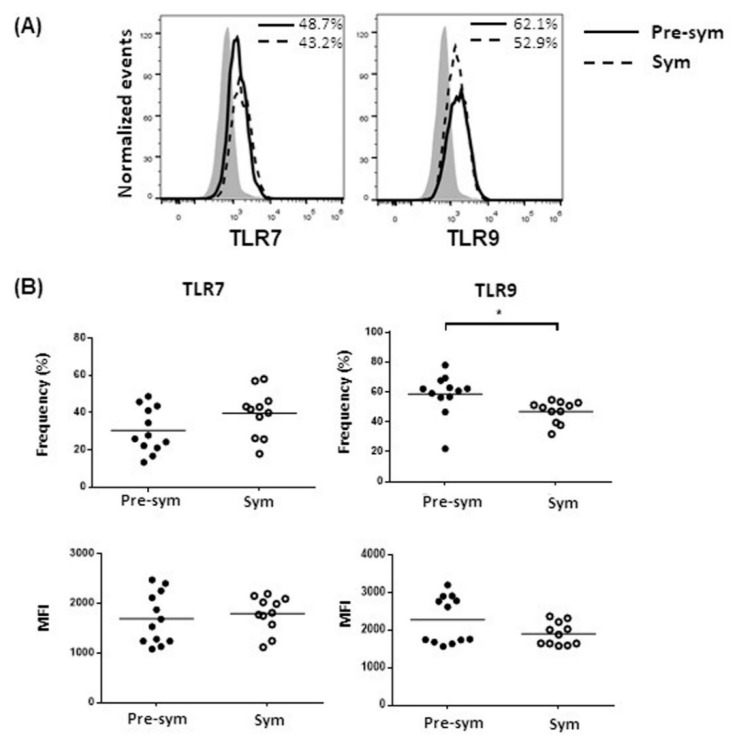
Symptomatic BWF1 CD11c^hi^CD11b^+^ DCs do not increase TLR7 and TLR9 expressions. Ex-vivo CD11c^hi^CD11b^+^ DCs in total splenocytes from pre-sym and sym BWF1 were stained and gated for CD11c^hi^CD11b^+^ cells to evaluate their intracellular TLR7 or TLR9 expressions. (**A**) Representative histograms showing the expressions of TLR7 and TLR9 in CD11c^hi^CD11b^+^, respectively, in pre-symptomatic (solid line) and symptomatic BWF1 (dotted line). Shaded histogram represents the isotype control. (**B**) Summary plots comparing TLR7 and TLR9 protein expressions in CD11c^hi^CD11b^+^ DCs from pre-sym and sym BWF1. Each symbol represents an individual mouse, and student’s t-test was used for statistical analysis (* *p* ≤ 0.05).

**Figure 6 ijms-20-04639-f006:**
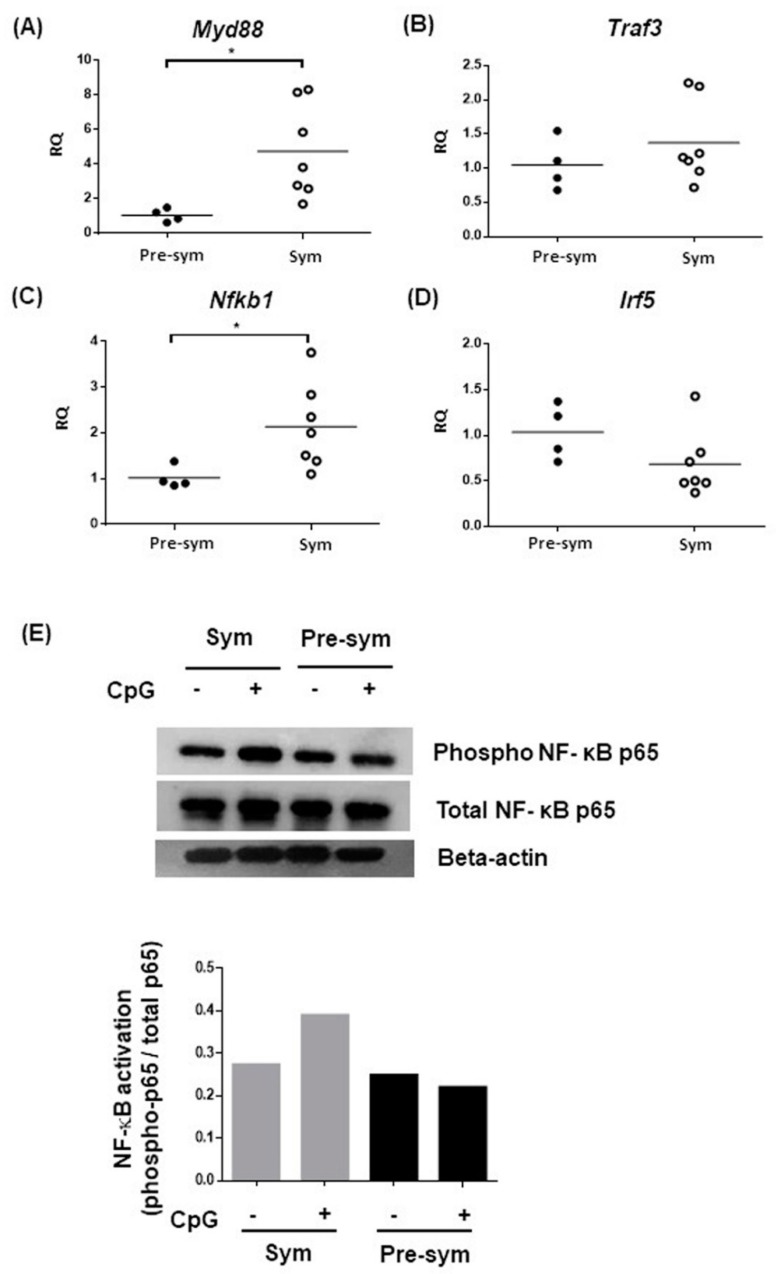
Expressions of different components of the TLR7 and TLR9 signaling pathway. Summary plots comparing the basal mRNA expressions of (**A**) *Myd88*, (**B**) *Traf3*, (**C**) *Nfkb1*, and (**D**) *Irf5*, in pre-sym and sym mice. Basal mRNA expressions were determined in purified CD11c^hi^CD11b^+^ DCs using qPCR. The mRNA level of each target was normalized with the house-keeping gene *β-actin*, and relative quantity (RQ) represents the mRNA level relative to CD11c^hi^CD11b^+^ DCs from pre-symptomatic BWF1. Each symbol represents an individual mouse, and student’s t-test was used for statistical analysis (* *p* ≤ 0.05). (**E**) FACS-purified CD11c^hi^CD11b^+^ DCs were stimulated with 5μM CpG for 20 min and assayed for NF-ҝB activation by western blotting analysis for phosphorylated p65. Twenty micrograms of protein samples were pooled from three independent stimulation experiments for one western blot. Densitometry analysis showed signal intensity of phosphorylated p65 normalized with total p65 of the respective samples.

**Table 1 ijms-20-04639-t001:** Primer sequence for qPCR.

Genes	Primer Sequence (Forward)	Primer Sequence (Reverse)	Product Size (bp)
*Il-6*	GAAGTTCCTCTCTGCAAGAGAC	CCAGAGAACATGTGTAATTAAGC	178
*Il-10*	GCCGGGAAGACAATAACTGC	TTCAGCTTCTCACCCAGGGA	280
*Tnfa*	AGCACAGAAAGCATGATCCGCG	AGAAGATGATCTGAGTGTGA	249
*Baff*	TCCAGCAGTTTCACAGCGAT	CCGGTGTCAGGAGTTTGACT	157
*Cxcl13*	CACGGTATTCTGGAAGCCCA	AGACAGACTTTTGCTTTGGACA	231
*Myd88*	TGTTCTTGAACCCTCGGACG	TTCTGGCAGTCCTCCTCGAT	241
*Traf3*	CAGCGCACGGCACAGAATTT	CTTGTAGCCTCCTTGCTCCG	149
*Irf5*	CTAGCCTGGATGTGGCATGT	CTCTTTAGCCCAGGCCTTGAA	314
*Nfkb1*	CCACAAGGGGACATGAAGCA	GATGGTACCCCCAGAGACCT	198
*β-actin*	AGATCAAGATCATTGCTCCTCCT	ACGCAGCTCAGTAACAGTCC	174

Forward and reverse primer sequence of each target is list in above. Efficiency of each primer set was validated and was comparable to the endogenous house-keeping gene *β-actin*.

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
