# Peer review of "Heightened TLR7/9-Induced IL-10 and CXCL13 Production with Dysregulated NF-ҝB Activation in CD11chiCD11b+ Dendritic Cells in NZB/W F1 Mice"

_ijms, 2019, doi:10.3390/ijms20184639_

Round 1
Reviewer 1 Report
The study by Yan et al., describes the role of TLR7/9 in mDCs regulate the SLE pathogenesis. In general the study is well performed and presented; there are two major draw back is in the study.
Fig 1. The authors are not given a serious thought on the mDC characterization and they used a general marker CD11c and CD11b for defining the mDCs. The marker can be expressed by the inflammatory DCs, subset of macrophages as well as mDCs. Also authors are not considering the role of the two subset of mDCs, the cDC1 ( CD8a/CD103) and cDC2 ( CD11b DCs). It is important to use additional markers such as XCR1/Clec9a/CADM1, SIRPA, CD206 etc to define the cell subset in the currently defined CD11c+CD11b mDC population. Please refer this article for the minimum number of markers to define mDCs or other DC subsets and mononuclear phagocytes (PMID:27637149). These markers can identify the subset specific role of mDCs or inflammatory DCs in CXCL13 production or the currently reported observation.
Figure 2 . The expression profile of CD80 and MHC II shows a bimodal histogram. Does it indicates one of the mDC subsets is down regulating the costimulatory molecules expression and other subset is not affected. It will be great if you can rule out any subset specificity on costimulatory molecule expression.
The other major draw back is the claim of elevated responses for TLR7/9 ligands: The authors should perform additional control with other TLR ligands like LPS and Poly I:C to confirm the specificity of TLR7/9 activation. The TLR3 in cDC1 can also respond strongly to the SLE and the role of TLR3 SNPs are reported in patients. We have to understand whether the mDCs in SLE model are preprogrammed to respond to any TLR activation or very specific to TLR7/9.
Author Response
We thank the reviewer’s constructive comments. Below is our specific reply and the corresponding changes in the revised manuscript are highlighted.
Our original attempt was to study mDCs as a whole population and we agree that the use of CD11c and CD11b were not sufficient to distinguish mDCs or conventional into cDC1 and cDC2 subsets, which have different phenotypic and functional properties. To better reflect this limitation, we have revised the text and the title, replacing mDCs to CD11chiCD11b+ DCs accordingly.To further address reviewer’s suggestion, we re-examined the expression of the Clec9A and CD172a in CD11chiCD11b+ splenic DCs in BWF1 mice. We observed that CD11chiCD11b+ DCs composed mainly cDC2 cells expressing CD172a but not Clec9A (marker of cDC1), and the frequency of CD172a+ cDC2 remained stable during lupus development. Furthermore, the downregulation of MHC-II and CD80 expression during lupus development were observed in these CD11chiCD11b+CD172a+ in BWF1 mice. We have added these additional information in Supplementary Figure 7 and added a new paragraph to discuss this (P.11-12) in the revised manuscript.
2. We agree that other TLR stimulations should be included as control to confirm specific heightened TLR7 and TLR9 responses. TLR4 response upon LPS stimulation had been performed and results showed no significant difference in Il-10 and Cxcl13 mRNA induction between pre-symptomatic and symptomatic BWF1. We have included these data in Supplementary Figure 5 and correspondingly added a sentence (P.5) in the revised manuscript.
Reviewer 2 Report
In their study Yim and co-workers compare the immuno-phenotype of splenic DC subpopulations derived from SLE-prone mice either prior to arisal of symptoms or after the onset of disease. The authors report that the freqeuncy of splenic pDC is reduced in case of illness whereas the fraction and overall number of mDC is elevated. mDC derived from ill mice were characterized by attenuated expression of MHCII and CD80 as compared with mDC obtained from healthy mice, but exerted similar allgoenic T cell activation. In addition, mDC of ill mice displayed higher expression of several inflammation markers (but also of IL-10) under basal conditions, and these DC were more responsive towards TLR9 stimulation than control DC, yielding elevated levels of IL-10 and CXCL13 protein. These differences were associated witgh higher MyD88/NF-kB1 mRNA expression at basal state, and somewhat higher NF-kB activation.
Major points:
1. a) Figure 1B: mDC comprise (at least) cDC1 and cDC2. Both subpopulations differ both in their immuno-phenotype and their functions. In their study, Yim et al. consider CD11c/CD11b double-positive cells as `mDC´. However, by that approach only cDC2 (and monocyte-derived/inflammatory DC) are detectable since cDC1 besides CD11c rather coexpress XCR1/CLEC9a and other markers than CD11b. The authors should either alter the text thoroughly to indicate so or perform adequate in depth flow cytometric analysis to substantiate their findings.
b) Figure 1C: The observation of higher levels of `mDC´ in symptomatic mice could be explained by inflammation-induced induction of `inflammatory DC´. The authors should test so by including more phenotypic markers or at least discuss this possibility.
2. Figure 2C: DC/T cell cocultures were performed at one ratio (1:10) only. However, in general the activity of different APC populations is assessed in cocultures at various APC/T cell ratios. First, this approach serves to rule out that in case of very potent antigen presenting cells (APC) the induced strong T cell proliferation is inhibited at the time point of 3H thymidine application due to limited nutrient supply under stationary conditions. (In that case APC with a very high and a rather moderate activity could actually yield similar T cell proliferation.) Second, functional differences between activated DC may become obvious rather at low DC:T cell ratios. The authors need to perform according assays.
3. Figure 6E: The authors state in the text (lines 204-206) that CpG treatment induced higher NF-kB phosphorylation in case of DC derived from symptomatic mice as compared to control DC. In the according graph no significancy is shown. The text needs to be adjusted. In addition, according to the figure legend "samples were pooled from 3 independent experiments". Therefore, it is not clear whether in each of the 3 experiments NF-kB activity was higher in `mDC´ of symptomatic mice. If the authors want to demonstrate increased NF-kB activity of `mDC´ derived from ill mice, this assay needs to be performed several times, and the derived data points need to be taken into account.
Minor points:
1. Figure 4: Stimulation of `mDC´ derived from healthy mice with CpG and R848 yielded no visible induction of IL-10 and CXCL13. The authors should either include a scale break to viusualize stimulation-dependent alterations in protein contents also in case of DC obtained from healthy mice or indicate in the text, e.g. in the figure legend, whether these DC expressed either protein in response to stimulation as well, albeit at lower level than DC derived from ill mice.
2. Figure 6A,C: The authors demonstrate that `mDC´ derived from ill mice express more MyD88 (4-fold) and NF-kB1 (2-fold) mRNA than control DC. The authors should perform intracellular flow cytometry or Western blots to confirm that the moderately elevated levels of MyD88/NF-kB1 mRNA were accompanied by higher protein contents.
Author Response
We thank the reviewer’s constructive comments. Below is our specific reply and the corresponding changes in the revised manuscript are highlighted.
Major points:
1a & b) Our original attempt was to study mDCs as a whole population and we agree that the use of CD11c and CD11b were not sufficient to distinguish mDCs or conventional into cDC1 and cDC2 subsets, which have different phenotypic and functional properties. To better reflect this limitation, we have revised the text and the title, replacing “mDCs” to “CD11chiCD11b+ DCs” accordingly.
To further address reviewer’s suggestion, we re-examined the expression of the Clec9A and CD172a in CD11chiCD11b+ splenic DCs in BWF1 mice. We observed that CD11chiCD11b+ DCs composed mainly cDC2 cells expressing CD172a but not Clec9A (marker of cDC1), and the frequency of CD172a+ cDC2 remained stable during lupus development. Furthermore, the downregulation of MHC-II and CD80 expression during lupus development were observed in these CD11chiCD11b+CD172a+ DCs in BWF1 mice. We have added these additional information in Supplementary Figure 7 and added a new paragraph to discuss this (P.11-12) in the revised manuscript.
2) We agree that titrating of DC/T cell ratio for the co-culture proliferation assay may yield more information and most studies use DC/T at ratio ranges from 1:10 to 1:1, harvesting at different time points. However, the abundance of CD11chiCD11b+ DCs, especially in the pre-symptomatic mice, was very low and imposed a major constrain for most of our functional experiments. It usually required at least five to seven pre-symptomatic mice for isolating CD11chiCD11b+ DCs for one single functional assay with duplicates or triplicates, and thus we could only choose the most appropriate condition. It has been shown that DC/T ratio of 1:10 can support proliferation whereas ratio at 1:2 induces proliferation arrest (PMID: 16180253), and we performed the proliferation assay at the lower end of the D/T ratio spectrum. To address this issue, we added a sentence (P.4) to address the limitation of our findings in this assay.
3) Similarly due to low DC abundance in pre-symptomatic mice, we had to pool cells from 3 independent stimulation experiments (each with 5-7 pre-symptomatic mice) in order to have sufficient protein for one western blot analysis, thus no error bar could be indicated. We understand the need to reproduce this observation independently, and thus recently resolve to use flow cytometry to detect NF-KB p65 phosphorylation in FAC-sorted CD11chiCD11b+ We observed similar pattern with higher induction of p65 phosphorylation in symptomatic mice when compared with pre-symptomatic mice (Supplementary Figure 6). Additional text to describe this result is highlighted (P.7).
Minor points:
Figure 4 legend title (P.7) has been modified to emphasize the absence of IL-10 and CXCL13 protein production upon TLR7/9 stimulation.
We don’t have data to show the increase of Myd88 and Nfkb1 mRNA was accompanied with evaluated protein expression in symptomatic CD11chighCD11b+ DCs due to limited material. We reason that functional deviation provides more prominent and direct evidence than expression differences to account for the heightened TLR7/9 response. Therefore, we provided additional data (major point 3) to demonstrate functional aberration of the NF-ҝB pathway. We sincerely hope the reviewer would take into the account of the constrains we have had in this study.
Round 2
Reviewer 1 Report
Yim et al., describes the role of CD11bhighCD11chigh cell subsets in SLE and potential mechanism of TLR7 & 9 in aggravating the problem. The authors improved the manuscript with additional experiments and using a general terminology to represent the mDC compartment. In general the manuscript is improved and some minor modifications may make it appear better. The modifications are suggested below.
Figure 1 A & B : It will be great if the authors can slightly modify the figures. Figure 1 A ( CD317 vs CD11c) clearly shows two populations of CD317+ ( gated as pDC and cross checked for Siglec H expression), and CD11c+. Please gate for the CD11c in the same dot plot and then generate a Dot plot for CD11c vs CD11b. Select the CD11bhigh as the population of your interest. Current gating doesn’t tell us where the cDC1 subset in your population. I agree with the additional staining confirming the SirpA expression and lack of Clec9a expression on the cell subset. It would have been nice to have CD206 staining to confirm the involvement of any inflammatory DCs, but the broad terminology CD11chigh CD11b high DC may be enough to convey the idea. Please add the p values for Supplementary Figure 5., visually the values have almost a two fold difference and it will be great to show the p value to convince the readers
thank you very much
Author Response
Point-to-point address to reviewer’s comments (Reviewer 1)
We appreciate the reviewer’s suggestion to improve our data presentation. The original experiments for Figure 1A and B were designed respectively for pDC and mDC staining and we did not include CD11b in pDC staining, thus we could not re-gate the data on CD11b for Figure 1A. It would be ideal to have additional markers to further delineate the location of cDC1 in our gating strategy and whether CD11c(hi) CD11b(+) gated DCs are inflammatory DCs; however, it is impossible for us to re-do the staining for so many animals at the moment in order to make a more specific claim. Nevertheless, we are glad that the reviewer agrees using the broader terminology CD11c(hi)CD11b(+) DC may be enough to convey our idea.
P-values and a statement on statistical analysis have been added in Supplementary Figure 5.